# Potentially Inappropriate Use of Opioids in the Management of Migraine in Colombia

**DOI:** 10.3390/biomedicines11092385

**Published:** 2023-08-25

**Authors:** Manuel E. Machado-Duque, Maria Fernanda Echeverry-Gutiérrez, Andrés Gaviria-Mendoza, Luis F. Valladales-Restrepo, Jorge E. Machado-Alba

**Affiliations:** 1Grupo de Investigación en Farmacoepidemiología y Farmacovigilancia, Universidad Tecnológica de Pereira-Audifarma S.A, Pereira 660003, Colombia; memachado@utp.edu.co (M.E.M.-D.); angaviria@utp.edu.co (A.G.-M.); lfvalladales@utp.edu.co (L.F.V.-R.); 2Grupo de Investigación Biomedicina, Facultad de Medicina, Fundación Universitaria Autónoma de Las Américas, Pereira 660003, Colombia; mafe.eg9@gmail.com

**Keywords:** opioid analgesics, migraine, drug-related side effects, adverse reactions, headache, pharmacoepidemiology

## Abstract

Research objective: To identify the frequency of opioid use in a group of patients diagnosed with migraine in Colombia. Methods: Study of a retrospective cohort of patients with a diagnosis of migraine and a first prescription of antimigraine drugs from emergency services and a priority outpatient clinic. Sociodemographic, clinical, and pharmacological variables were identified; a 12-month follow-up was carried out to identify the use of a new opioid. Results: A total of 6309 patients with a diagnosis of migraine were identified, with a mean age of 35.5 ± 12.3 years, of which 81.3% were women. Nonsteroidal anti-inflammatory drugs (51.1%) were the most frequently prescribed medications, followed by ergotamine + caffeine (31.3%), acetaminophen (15.05%), and acetaminophen + codeine (14.4%). At the time of the index, 1300 (20.6%) patients received some opioid. During the follow-up, a total of 1437 (22.8%) patients received a new opioid, of which 31.8% belonged to the group that received an initial opioid and 20.4% to the group that did not receive one, which was statistically significant (OR:1.81; 95%CI:1.58–2.07; *p* < 0.001). Conclusions: The frequent use of opioids in the management of migraines is potentially inappropriate and can lead to problems of tolerance, abuse and dependence. This combined with the low prescription of triptans, offers an opportunity for improvements in medical practice.

## 1. Introduction

Migraine is classified as one of the most prevalent chronic disorders in the world. According to the World Health Organization, it affects approximately 15% of the population globally and represents a fifth of neurological consultations [1].

The American Neurological Association makes explicit recommendations, with multiple therapeutic options that are low-cost and adaptable to the individual patient. In the case of acute management for migraine attacks, the use of nonsteroidal anti-inflammatory drugs (NSAIDs), including aspirin, nonopioid analgesics, acetaminophen or combinations of analgesics with caffeine, is proposed for mild to moderate intensity, and specific agents such as triptans for moderate to severe intensity. In addition, the latter are recommended in mild to moderate episodes that do not respond to initial management with NSAIDs or combinations with caffeine [2]. However, Lim et al., published a systematic review in the Journal of Clinical Medicine, in which emergency department prescription patterns of medications for acute migraine were evaluated. A tendency to underuse triptans and overuse opioids was observed; these are behaviors not based on evidence [3].

The national and international literature presents gaps related to the prescription of opioids and their adverse effects when they are used to manage migraine. Although opioids are not indicated for their treatment, in Colombia, for example, there is a considerable number of patients treated with this group of drugs, as shown in a 2020 study. This study showed the prescription patterns of drugs used for the management of headaches (including migraine) and evaluated these in a population of approximately 6.5 million people. This exposed the use of opioids in 23.5% of the cases, with the most commonly used combination being acetaminophen plus tramadol [4]. This prescription is controversial, since opioids can perpetuate crises, lead to hyperalgesia, tolerance and abuse disorders, as well as chronic migraine; while doctors consider their analgesic efficacy, and that they do not contribute to the pathophysiological solution of the disorder, they should also ensure that these drugs do not increase patient problems [3,5,6].

The constant increase in the prescription and use of opioids worldwide has motivated studies that estimate adverse effects triggered by the pharmacological prescription of these, such as dependence and chronic pain, among others. This has been shown in the US, where 2 million people have opioid use disorder, which represents an estimated economic cost of USD 78.5 billion per year and a significant dependence of the population on this group of drugs [7]. Therefore, demonstrating the evidence of potential errors and promoting to physicians the adjustment of therapies and knowledge linked to scientific evidence, specifically in the treatment of migraine, may have a positive impact on the effectiveness, safety and costs of the therapy [2,8].

It is also important to highlight that some studies [9,10,11], including one in Colombia, have shown how the use of an opioid, such as tramadol, in emergencies increased the probability of receiving a new prescription in the following 12 months. This in turn increased the risk of adverse reactions and related problems [11] to these drugs, for example, in patients with migraine in which their use is not recommended. For this reason, the development of studies that show the real-life use of drugs, both those indicated and those without evidence of use, or that their risk–benefit balance is inadequate, can support the identification of problems related to prescription. In addition they can support the design of improvement programs in prescription by doctors, and benefit patients.

The Colombian health system offers coverage to the entire population through two regimes, one contributory or paid by employers and workers, and another subsidized by the state. The state’s insurance has a benefit plan that includes painkillers, such as antimigraine medications and opioids. Because migraine and the prescription of opioids in those who suffer from them are two of the problems raised to be intervenable, real estimates are required at the national level to know and understand the frequency of the disorder, and associated factors that can contribute to the appropriate treatment. For this reason, the objective of identifying the frequency of opioid use in a group of patients from Colombia with a diagnosis of migraine was proposed, including treatment patterns during emergency management and priority consultation, as well as new opioid prescriptions in the subsequent 12 months and the factors associated with these prescriptions.

## 2. Materials and Methods

### 2.1. Study Design and Settings

An observational study of a retrospective cohort was developed, which included patients with a diagnosis of migraine, according to the code in the International Classification of Diseases (ICD-10), registered in the database of drug dispensing of Audifarma S.A., Pereira, Colombia. The Colombian health system provides universal coverage for access to health services, through two schemes, one subsidized for those without work or with low economic resources, and another supported by contributions from workers and companies. Each person is affiliated with an insurer, which, at the time of receiving a prescription for a medication from a patient, goes to a logistics operator in charge of dispensing the medication. At that time, the delivered drugs, and other clinical and sociodemographic variables are recorded in the drug claim database.

We included patients with a diagnosis of migraine older than 14 years of age and of any sex, residing in one of the cities according to regions of the country, with a first prescription of migraine medications during the last quarter of 2020 from emergency services and priority external consultations, and registered in the database at least 12 months before the index date. Patients with opioid use in the six months prior to the inclusion index date, those with diagnoses of pathologies with indication of chronic opioid use (cancer, acute and severe pain, in palliative care) and patients with incomplete information were excluded.

For each selected patient, different variables of interest were evaluated during the subsequent 12 months to the index date. The information was obtained from the drug dispensing database of Audifarma S.A and a dataset with the registered information of each patient was constructed. This database included the population from all regions of the country, and from all socioeconomic strata, affiliated with the contributory and subsidized regimes of the health system. The time of the first prescription of opioids or other medications for the management of migraine during the observation period was determined as the index date. Follow-up was performed for 12 months from this date and continued until a patient had a new prescription for an opioid or until 31 December 2021. The following variables were identified:Sociodemographic: age (years), sex (female, male), geographic region of residence according to the National Administrative Department of Statistics (DANE) of Colombia, insurer;Diagnosis: migraine (ICD-10 code identified; G43.0 migraine without aura [common migraine]; G43.1 migraine with aura [classic migraine]; G43.2 migraine state; G43;Complicated migraine; G43.8 other migraines; G43.9 migraine, unspecified), date of first migraine diagnosis (index date);Pharmacological: The medications used for the management of migraine were identified by classifying them into the following groups: acute episode, antimigraine (triptans, ergot derivatives), analgesics (acetaminophen, NSAIDs), and other pain relievers (opioids, metoclopramide);Use of opioids: In those patients who identified the use of opioids in the management of migraine, a comparison was made with those who did not receive opioids on the index date. During the subsequent 12 months, patients were monitored to determine the frequency of new opioid prescriptions, and the specialty of the prescribing physician was identified. The time of the first prescription of opioids or other medications for the management of migraine during the observation period was determined as the index date. Follow-up was performed for 12 months from this date and continued until a patient had a new prescription for an opioid or until 31 December 2021.

### 2.2. Bioethical Considerations

The protocol received the endorsement of the Bioethics committee of the Universidad Tecnológica de Pereira in the category of research without risk (approval code: 70-060921, 6 September 2021). The ethical principles established by the Declaration of Helsinki were respected.

### 2.3. Statistical Analysis

The analyses were carried out using the statistical software SPSS 26.0 for Windows (IBM Corp., Armonk, NY, USA). Univariate analyses were performed with frequencies and proportions for categorical variables, and descriptions of measures of central tendency for continuous variables. Bivariate analyses were performed using the *χ*^2^ test to identify significant differences between covariates and new opioid use during follow-up. A multivariate logistic regression model was constructed, seeking to adjust the comparisons of the covariates and the new use of opioids in patients treated for migraine, including initially associated variables in the dataset through bivariate analyses, as well as those with biological plausibility to explain the outcome. A value of *p* < 0.05 was considered significant.

## 3. Results

A total of 6309 patients with a diagnosis of migraine were included during the months of October, November and December 2020, with a mean age of 35.5 ± 12.3 years (median of 33.8 years and interquartile range 25.8–43.1 years) and a predominance of females (*n* = 5131; 81.3%). No statistically significant differences were found between the mean age in males and females (*p* = 0.47). The patients resided mainly in the Caribbean region (*n* = 2166; 34.3%), followed by the Central region (*n* = 1853; 29.4%), Bogotá-Cundinamarca region (*n* = 123; 28.9%), Pacific region (*n* = 302; 4.8%) and Eastern region (*n* = 165; 2.6%). Table 1 shows migraine diagnosis and sociodemographic characteristics.

The drugs most frequently used in patients with a diagnosis of migraine were, with at least one prescription, NSAIDs (*n* = 3224; 51.1%), followed by specific antimigraine drugs, such as ergotamine + caffeine (31.3%) and acetaminophen. In addition, the results found that 1300 (20.6%) patients received some type of opioid in the same period of time. Table 2 shows the frequencies of use for each medication.

The follow-up revealed that 1437 (22.8%) patients received a new opioid within 12 months of the index date. Figure 1 shows the patient inclusion flowchart. In the group that received an opioid first, a total of 413 (31.8%) received another opioid in the following year, whilst 1024 (20.4%) patients who did not receive an opioid during the index period did receive opioids during their follow-up. This difference was statistically significant (OR:1.81; 95%CI:1.58–2.07; *p* < 0.001).

During the 12-month observation, the most commonly used medication was acetaminophen/codeine (*n* = 1080; 75.2%), followed by tramadol (*n* = 446; 31.0%), morphine (*n* = 24, 1.7%), meperidine (21, 1.5%), oxycodone (*n* = 6, 0.4%) and diclofenac/codeine (*n* = 1, 0.07%). The medications were prescribed by physicians, including 6030 (95.6%) formulations made by general practitioners, followed by general surgeons with 83 (1.3%), internists with 50 (0.8%) and 146 (2.3%) others.

The multivariate analysis that considered the risk of receiving an opioid up to 12 months after care for a migraine attack found that having received an opioid analgesic at the index moment or an antiemetic, being treated in the Caribbean region and being age 30 or older made individuals statistically more likely to receive an opioid during follow-up, while those treated in the Eastern region showed a lower risk (see Table 3).

## 4. Discussion

The different clinical practice guidelines and consensus regarding the management of acute migraine episodes recommend avoiding the use of opioids as analgesics. However, as evidenced in this study in Colombia, opioids are widely prescribed for this indication in more than a fifth of patients. This is despite the fact they may have limited effectiveness and increase the risk of overuse and chronification of migraine, as well as creating the possibility of abuse and dependence [5,12].

The use of ergotamine + caffeine more frequently than triptans does not comply with the recommendations of Colombian experts [13] and international guidelines for the management of acute migraine episodes [2]. This is mainly due to safety problems derived from its low selectivity for the serotonin receptor, which increases the probability of adverse cardiovascular and neurological reactions [14,15]. In Colombia, the greater use of ergotamine was also associated with a poor recommendation for dosage and interval in almost all patients to whom it was prescribed [16]. This increases the probability of adverse reactions; added to the inappropriate use of opioids, this can be associated with the chronification of migraine [17]. Publications have shown in the United States that more than 50% of patients receive triptans [18], compared to less than 5% in this study, even though the medication is covered by the Colombian Health System [19]. This situation can be explained through difficulties in access to the medication or barriers to their prescription by the insurers of the patients, or ignorance of the doctor regarding the current recommendations for management versus effectiveness and safety in the use of the migraineurs.

The most widely used opioids for pain management in this group of patients were codeine + acetaminophen and tramadol, which have already been previously described in studies in Colombia [20,21]. This may be explained by doctors’ ignorance about the effectiveness of various antimigraine medications and analgesics, accompanied by a feeling of greater confidence in the use of opioids for any clinical condition that causes pain [11,14,22]. The relationship between the use of opioids and some negative outcomes in patients with migraine, such as risk of overuse, chronicity of migraine and limited effectiveness, is clear [5]. This creates an opportunity to improve the use of migraine control medications based on the best scientific evidence available [1,2]. Reports from other countries have also found a high frequency of opioid use, as shown by the CaMEO study in the United States (36.3% of cases) [23] and by Gunasekera et al., in Australia (50% of patients seen in emergency services) [22], which shows that this practice is widespread. The use of opioids in patients with migraine episodes can be explained by ignorance of the correct management recommendations, which advise against their use [5], or by an incorrect diagnosis related to well-defined symptoms, lack of clarity about the pathophysiological process that triggers the migraine, and even the severity of the crisis [2].

The most relevant finding in this analysis was the identification of a 64% higher adjusted probability of receiving an opioid again in the 12 months following its first prescription. This is consistent with other studies that have reported an increase in this risk after a first prescription, not only in migraine [23,24,25], but also during consultations for pain of other origins in emergency services and general care settings [11]. The prudent use of opioids is of great relevance in indications in which they are clearly recommended, especially for the management of migraine, since they have been associated with chronic use [26].

The identification of the increased probability of receiving opioids as age increases is consistent with the findings of other studies [11,25,27] and generates an additional concern, since it is the elderly who are more prone to adverse reactions such as drowsiness, falls with hip fracture and delirium [5]. In this analysis, no relationship was found with the sex of the patients, but it should be taken into account that other studies have found that being femalewas related to a greater probability of receiving opioids [25,28].

Finally, it was identified that almost all of the prescriptions were made by general practitioners, which provides an opportunity to implement continuing education and therapeutic update strategies for this group of professionals, in an effort to reduce potentially inappropriate opioid prescriptions. The findings in differences between regions have been previously described in pharmacoepidemiological studies, opening opportunities for the implementation of clinical practice guidelines and updates for physicians, especially those in regions with greater formulations [4,16,21].

Some limitations of the study are inherent to the source of information; in particular, dispensations linked to the diagnoses did not record the medical formulas, and there was no data in the medical records, images or laboratory results. In addition, there were difficulties in being able to clarify the criteria used to diagnose migraine and which professional made the diagnosis. Pain severity was measured at the time of consultation, which has been related to the probability of prescription of opioids [22]. It is possible that the true diagnosis of migraine was not represented by the ICD-10 code registered for the prescription, and the use of opioids in emergency services and priority consultations could be even higher. It was not possible to establish or compare whether there was opioid abuse according to the different age groups or the sex of the patients. Furthermore, it is not possible to recognize the use of drugs purchased outside the health system, and it is considered that the findings should only be extrapolated to populations with similar insurance characteristics, because the population included is that of the insurers to which medicines are delivered by Audifarma S.A, and there may be some differences with the general population. Some strengths are recognized, especially the number of patients who are part of the database and the large number of subjects with a diagnosis of migraine included in the analysis, as well as the rigor of the 12-month follow-up to identify subsequent uses of opioids.

## 5. Conclusions

With the above findings, it can be concluded that most individuals who seek medical care for migraine episodes are women who receive NSAIDs, ergotamine + caffeine and opioids. Those patients who receive an analgesic have a greater probability than those treated with other painkillers of receiving an opioid in the following 12 months, which increases the risk of adverse effects. It is necessary to continue examining the associations between the use of opioids and the risks to the health of patients, and to issue clear recommendations that are adopted by physicians to guarantee care that offers the best quality and safety. In addition, it is very important to make the treating physicians aware of the risks that are generated when using opioids in their patients who present with migraine, especially the risk of tolerance, abuse and dependency. The limited use of triptans offers an opportunity to improve care with drugs that have been shown to be effective.

## Figures and Tables

**Figure 1 biomedicines-11-02385-f001:**
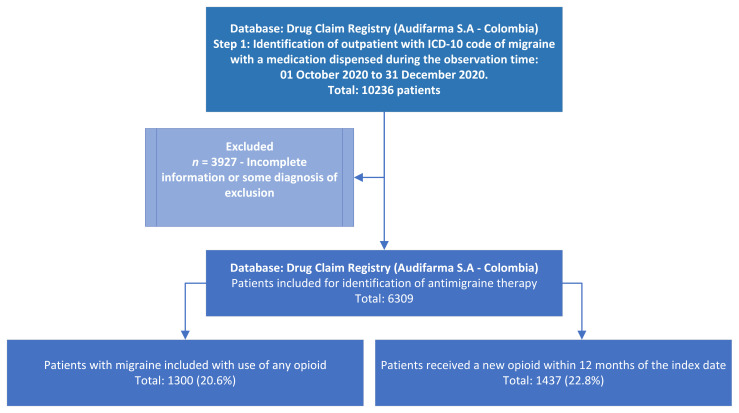
Patient inclusion flowchart.

**Table 1 biomedicines-11-02385-t001:** Sociodemographic and admission diagnoses in a group of patients diagnosed with migraine affiliated with the Colombian Health System, 2020.

Variable	Frequency	%
Sociodemographic		
Female—No. (%)	5131	81.3
Age in years (Mean, SD)	35.5 ± 12.3	
Male–Female age (mean, SD)	35.2 (12.7)–35.5 (12.1)	
Age Group—No.		
Age 14–30 years	2425	38.4
Age 30–45 years	2540	40.3
Age 45–60 years	1095	17.4
Age > 60 years	249	3.9
Admission diagnoses		
Migraine, unspecified	3633	57.6
Migraine without aura	807	12.8
Migraine with aura [classical migraine]	606	9.6
Status migrainosus	481	7.6
Complicated migraine	401	6.4
Other migraine	381	6.0

**Table 2 biomedicines-11-02385-t002:** Frequency of use of drugs to control acute attacks in a group of patients diagnosed with migraine affiliated with the Colombian Health System, 2020.

Medication Used	Frequency	%	Most Used Presentation	Dose (Median)	DDD *
Non-specific analgesics					
Naproxen	1823	28.9	250 mg tablet	750 mg	1.5
Acetaminophen	946	15.0	500 mg tablet	1500 mg	0.5
Diclofenac (ampule)	806	12.8	75 mg amp	75 mg	0.75
Dipyrone	401	6.4	1 gr amp	1 g	0.33
Ibuprofen	355	5.6	400 mg tablet	1200 mg	1.0
Diclofenac (tablet)	168	2.7	50 mg tablet	125 mg	1.25
Celecoxib	1	0.02	200 mg tablet	200 mg	1.0
Opioids					
Tramadol	433	6.9	50 mg amp	40 mg	0.13
Meperidine	6	0.1	100 mg amp	100 mg	0.25
Morphine	2	0.03	10 mg tablet	10 mg	0.33
Oxycodone	1	0.02	10 mg tablet	20 mg	0.26
Acetaminophen/Codeine	908	14.4	Tablet 325 + 8 mg	650 + 16 mg	0.2 + 1
Specific antimigraine drugs					
Ergotamine + caffeine	1972	31.3	Tablet 1 + 100 mg	1 mg	0.25
Naratriptan	125	2.0	2.5 mg tablet	2.5 mg	1.0
Sumatriptan	103	1.6	50 mg tablet	50 mg	1.0
Zolmitriptan	14	0.2	Nasal Sol. 5 mg/dose	7 mg	2.8
Eletriptan	2	0.03	40 mg tablet	40 mg	1.0
Antiemetics					
Metoclopramide	731	11.6	10 mg tablet	10 mg	0.33
Ondansetron	42	0.7	Amp 4 mg	4 mg	0.25

* relationship between mean dose and defined daily dose.

**Table 3 biomedicines-11-02385-t003:** Multivariate Logistic Regression, evaluating association with opioid prescription during a 12-month follow-up in patients with migraine from Colombia.

Variable	*p* Value	OR Adjusted	95% Confidence Interval	
			Lower	Upper
Opioid prescription (index)	<0.001	1.65	1.41	1.93
Acetaminophen (index)	0.932	0.99	0.83	1.19
NSAIDs (index)	0.850	1.01	0.88	1.16
Dipyrone (index)	0.168	1.19	0.93	1.53
Ergotamine + caffeine (index)	0.953	1.00	0.86	1.17
Triptans (index)	0.714	1.04	0.84	1.29
Antiemetics (index)	<0.001	1.40	1.16	1.68
Female gender	0.164	1.12	0.96	1.31
Age 14–30 years	Reference			
Age 30–45 years	<0.001	1.56	1.36	1.79
Age 45–60 years	<0.001	1.70	1.43	2.02
Age > 60 years	<0.001	1.97	1.46	2.66
Bogota-Cundinamarca Region	Reference			
The Caribbean Region	<0.001	1.33	1.14	1.55
Central Region	0.621	1.04	0.88	1.23
Eastern Region	0.003	0.46	0.28	0.77
Pacific region	0.933	0.99	0.72	1.34

## Data Availability

The data are available on: https://www.protocols.io/private/00FD035DB90911EDAAE90A58A9FEAC02 (accessed on 20 August 2023).

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
