# Peer review of "Potentially Inappropriate Use of Opioids in the Management of Migraine in Colombia"

_biomedicines, 2023, doi:10.3390/biomedicines11092385_

Round 1
Reviewer 1 Report
Dear Authors,
I have read your paper with interest. Malpractice in therapy of migraine is important as it may lead to exacerbation of migraine and its transformation to chronic forms of headache. That is why your paper is important.
I have some suggestions to improve the paper
1. Can you describe patients with migraine in a more detailed way, showing distribution of various forms of migraine (according to ICHD or ICD10)?
2. Most commonly used opiod was codeine. It is a mild opioid. Please, provide some clinical/epidemiological data on negative consequences of using specifically codeine in therapy of migraine.
3. Can you get some data about frequency of prescribing each of the drug? It would add some information about intensity of consumption of opioids.
4. Can you provide any data how frequently patients were seeking for medical help due to migraine, depending on drug they were prescribed?
Quality of English is fine.
Author Response
Manuscript ID: biomedicines-2525780
Type of manuscript: Article
Title: Potentially inappropriate use of opioids in the management of
migraine in Colombia
Dear reviewer 1
We appreciate your comments. We are sure that the observations made by everyone contribute to improving the quality of the manuscript.
I have read your paper with interest. Malpractice in therapy of migraine is important as it may lead to exacerbation of migraine and its transformation to chronic forms of headache. That is why your paper is important.
I have some suggestions to improve the paper
- Can you describe patients with migraine in a more detailed way, showing distribution of various forms of migraine (according to ICHD or ICD10)?
A/ A new table 1 is added, with information on the diagnostic distribution of migraine.
- Most commonly used opioid was codeine. It is a mild opioid. Please, provide some clinical/epidemiological data on the negative consequences of using specifically codeine in therapy of migraine.
A/ In the literature, opioids are included as a group, and their consequences such as adverse reactions and complications are considered grouped. There are no specific studies with the following search (codeine AND migraine) on pubmed.
- Can you get some data about the frequency of prescribing each of the drug? It would add some information about intensity of consumption of opioids.
A/ Based on the study design, we did not seek to identify the duration of treatment during the management of the episode in the emergency room or priority care. Neither was the intensity of use in follow-up. It was identified if the included patient received it, at what dose, and which one he used.
- Can you provide any data how frequently patients were seeking for medical help due to migraine, depending on drug they were prescribed?
A/ To answer this question, it would be necessary to carry out the study with another objective, looking for the frequency of medical consultation with the emergency services. Being a different objective than the one set. With the information collected for the primary objective, we cannot answer the frequency of consultations for migraine.
The authors
Reviewer 2 Report
The problem of primary headache therapy is relevant in modern neurology, despite the development and registration of next-generation anti-migraine drugs. The prescription of opioids is a significant problem for health care in many regions of the world. This explains the relevance of the study. However, the manuscript needs a serious revision.
Major comments:
The design of the study is weak. No comparison groups have been formed. How was the sample size calculated?
Minor comments:
Lines 6-12: Please write in English.
Line 14: Change "Background" to "Research Objective".
Line 37: Remove the extra parentheses.
Line 53: Delete "+", write "and".
Line 59: Add links to research after the phrase "some studies", as well as a link after the phrase "one in Colombia".
Subsection 1.2: Add a flowchart of the design of your study. Who diagnosed migraine? A neurologist? A general practitioner?
The criteria for the diagnosis of migraine presented in the 3rd edition of the International Classification of Headache were used?
Line 111: Use only the abbreviation «NSAID», because you have used it earlier in the Introduction section.
Subsection 2.2.: Your study included not only adults, but also adolescents. Who signed the voluntary informed consent? Patients or their legal representatives (parents, guardians)? Please describe it in more detail.
Did patients receive remuneration for participating in this study? Please add this information.
Add the date of the Ethics Committee protocol (date, month, year).
Line 127: Delete the colon at the end of the subsection name.
Line 140 et seq.: replace "average" with "mean".
Results Section: Add mean age for female and male. Were there statistically significant differences in the study participants depending on their gender?
The sample was heterogeneous by gender of participants. Why weren't the male ones deleted? This could minimize the errors.
Was the sample homogeneous by age? Please add a graph of the distribution of samples by age or add the median age, 25 and 75 quartiles (for the total sample, female and male).
Are opioids prescribed by a doctor in Colombia? Who and why prescribed opioids to patients more often? Was it a consequence of insufficient training of doctors? Or were there other reasons? Which ones?
Add the Limitations section after the Discussion section. Please describe in detail the limitations of this study.
The References section needs technical correction.
The style of the English language needs a serious revision. The sentences are very long. It also makes it difficult to read the manuscript.
The English language needs a serious revision.
Author Response
Manuscript ID: biomedicines-2525780
Type of manuscript: Article
Title: Potentially inappropriate use of opioids in the management of
migraine in Colombia
Dear reviewer 2
We appreciate your comments. We are sure that the observations made by everyone contribute to improving the quality of the manuscript.
The problem of primary headache therapy is relevant in modern neurology, despite the development and registration of next-generation anti-migraine drugs. The prescription of opioids is a significant problem for health care in many regions of the world. This explains the relevance of the study. However, the manuscript needs a serious revision.
Major comments:
The design of the study is weak. No comparison groups have been formed. How was the sample size calculated?
A/ The study design was not intended to include two or more groups. We only had one cohort or group of patients. We have adjusted the first sentence of ‘Materials and Methods’ to indicate this. We did not calculate a sample size, all patients that fulfilled inclusion criteria were analyzed.
Minor comments:
Lines 6-12: Please write in English.
A/ These are proper names, with which groups and institutions are recognized, for which we consider they should not be translated and in all the journals where we publish we keep the name in its original language.
Line 14: Change "Background" to "Research Objective".
A/ It is changed in the text
Line 37: Remove the extra parentheses.
A/ It is removed from the text
Line 53: Delete "+", write "and".
A/, fits in text
Line 59: Add links to research after the phrase "some studies", as well as a link after the phrase "one in Colombia".
A/ New references are added. The one from Colombia It is reference 9. It is included at the end of the complete sentence.
Subsection 1.2: Add a flowchart of the design of your study. Who diagnosed migraine? A neurologist? A general practitioner?
A/ The flowchart is figure 1 with the design and selection. The diagnosis is the one registered in the drug database. It could be performed by any medical specialty that issued a medical prescription associated with the ICD-10 diagnosis.
The criteria for the diagnosis of migraine presented in the 3rd edition of the International Classification of Headache were used?
A/We could not confirm exactly what criteria each doctor used to diagnose migraine, but it is most likely that they use the diagnostic standard which is the 3rd edition of the International Classification of Headache.
Line 111: Use only the abbreviation «NSAID», because you have used it earlier in the Introduction section.
A/ fits alone with NSAIDs
Subsection 2.2.: Your study included not only adults, but also adolescents. Who signed the voluntary informed consent? Patients or their legal representatives (parents, guardians)? Please describe it in more detail.
A/ none, it is a retrospective investigation, with all the information collected on a database of drug claims. It is also classified as risk-free research. Finally. The bioethics committee exempts from the need for informed consent due to the type of study and the source of research data.
Did patients receive remuneration for participating in this study? Please add this information.
A/ As explained in the methods, it is a study carried out in a database. Specifically one related to drug claims. There was no contact with the patients.
Add the date of the Ethics Committee protocol (date, month, year).
A/ is included.
Line 127: Delete the colon at the end of the subsection name.
A/ deleted
Line 140 et seq.: replace "average" with "mean".
A/ changed.
Results Section: Add mean age for female and male. Were there statistically significant differences in the study participants depending on their gender?
A/ Both with a mean of 35 years. There are no significant differences, it includes: No statistically significant differences were found between the mean age in men and women (p=0.47)
The sample was heterogeneous by gender of participants. Why weren't the male ones deleted? This could minimize the errors.
A/ The idea of the study is to show the evidence in the real world, regarding migraine management and opioid prescription in this condition.
Was the sample homogeneous by age? Please add a graph of the distribution of samples by age or add the median age, 25 and 75 quartiles (for the total sample, female and male).
R/ the sample is homogeneous, by age between the sexes, median and interquartile range are included. A new table 1 is created
Are opioids prescribed by a doctor in Colombia? Who and why prescribed opioids to patients more often? Was it a consequence of insufficient training of doctors? Or were there other reasons? Which ones?
A/ Yes, the opioids are prescribed by doctors in Colombia. There are many reasons, from ignorance regarding management recommendations, pressure from the patient. It is a worldwide phenomenon in the face of prescription errors and inappropriate use of opioids. Line 253 – 259 talks about possible explanations with doctors and possibilities for improvement
Add the Limitations section after the Discussion section. Please describe in detail the limitations of this study.
A/ There is a full discussion section where the limitations are included and explained.
The References section needs technical correction.
The style of the English language needs a serious revision. The sentences are very long. It also makes it difficult to read the manuscript.
A/ Certificate of American Journals Experts is sent
The authors
Reviewer 3 Report
This is a helpful addition to the literature.
I note that the information was obtained from a database: "The information was obtained from the drug dispensing database of Audifarma SA." It is not made clear how a person in Colombia gets onto this database. Are all patient visits recorded or only those where a prescription is written? Are all people in the country recorded or only those who insure with a particular insurer, or those who use medications from a particular provider? If inclusion in the database is not universal, some explanation about potential biases should be included: for example are those in the database likely to be of higher or lower socio-economic group than those who are not?
On line 163, "The medications were prescribed by specialists ... ". This seems to include GPs who would not be regarded as specialists by at least some readers. I suggest omitting or clarifying this potentially misunderstood reference to specialists.
English is mostly fine.
Author Response
Manuscript ID: biomedicines-2525780
Type of manuscript: Article
Title: Potentially inappropriate use of opioids in the management of
migraine in Colombia
Dear reviewer 3
We appreciate your comments. We are sure that the observations made by everyone contribute to improving the quality of the manuscript.
This is a helpful addition to the literature.
I note that the information was obtained from a database: "The information was obtained from the drug dispensing database of Audifarma SA." It is not made clear how a person in Colombia gets onto this database.
A/ The Colombian health system provides universal coverage for access to health services, through two schemes, one subsidized for those without work or low economic resources and another supported by contributions from workers and companies. Each person is affiliated with an insurer, which, at the time of receiving a prescription for a medication from a patient, goes to a logistics operator in charge of dispensing the medication. At that time, the drugs delivered and other clinical and sociodemographic variables are recorded in the drug claim database.
Are all patient visits recorded or only those where a prescription is written?
A/ Only those with prescription medication and dispensing.
Are all people in the country recorded or only those who insure with a particular insurer, or those who use medications from a particular provider?
A/ Those registered with certain insurers, who contract the dispensing process with Audifarma S.A.
If inclusion in the database is not universal, some explanation about potential biases should be included: for example are those in the database likely to be of higher or lower socio-economic group than those who are not?
A/ The population included is that of the insurers to which medicines are delivered by Audifarma S.A, for which there may be some differences with the general population. We add in methods “This database includes population from all regions of the country, from all socioeconomic strata, affiliated with the contributory and subsidized regimes of the health system”.
On line 163, "The medications were prescribed by specialists ... ". This seems to include GPs who would not be regarded as specialists by at least some readers. I suggest omitting or clarifying this potentially misunderstood reference to specialists.
A/ term is adjusted
The authors
Round 2
Reviewer 1 Report
Dear Authors,
THank you for revision.
I have no further comments.
Author Response
Thank you
Reviewer 2 Report
The authors have improved their manuscript somewhat, but the Limitations section has not been highlighted. This is important because in addition to the Limitations section of this study, which the authors mentioned in the Discussion section, there are other important limitations that the authors indicated when responding to my comments. For example, difficulties in clarifying on the basis of which criteria and who diagnosed migraines; difficulties in comparing adherence to opioid abuse in different age groups and depending on the gender of patients; etc.
The "Discussion" section seems to me to be the weak point of this manuscript.
I still think this manuscript is weak, but I leave the solution of this issue to the Academic Editor of the journal.
Minor comments:
Lines 7-12 - the names of the affiliations have not been translated into English.
Line 20 - in the abstract, write "nonsteroidal anti-inflammatory drugs", but not “NSAIDs”.
Line 28 - separate the two keywords "drug-related side effects" and "adverse reactions".
Line 33 - the WHO abbreviation is used only once in the text, so it is better to delete this abbreviation; please do not use abbreviations if used three or less times.
Line 166 - replace "men and women" with "male and female", because teenagers were included in the study.
Line 185 - remove the bold in the text.
Author Response
Manuscript ID: biomedicines-2525780
Type of manuscript: Article
Title: Potentially inappropriate use of opioids in the management of
migraine in Colombia
Dear reviewer 2
We try to respond to each of your concerns and comments.
The authors have improved their manuscript somewhat, but the Limitations section has not been highlighted. This is important because in addition to the Limitations section of this study, which the authors mentioned in the Discussion section, there are other important limitations that the authors indicated when responding to my comments. For example, difficulties in clarifying on the basis of which criteria and who diagnosed migraines; difficulties in comparing adherence to opioid abuse in different age groups and depending on the gender of patients; etc.
The "Discussion" section seems to me to be the weak point of this manuscript.
I still think this manuscript is weak, but I leave the solution of this issue to the Academic Editor of the journal.
Response/ we made recommended adjustments for study limitations. And add some comments in conclusions.
Minor comments:
Lines 7-12 - the names of the affiliations have not been translated into English.
Response/ The names of the research group and the universities are proper names and should not be translated. This group has published all their research under the name in the original language. Thanks for understanding.
Line 20 - in the abstract, write "nonsteroidal anti-inflammatory drugs", but not “NSAIDs”.
Response/ is corrected.
Line 28 - separate the two keywords "drug-related side effects" and "adverse reactions".
Response/ is corrected.
Line 33 - the WHO abbreviation is used only once in the text, so it is better to delete this abbreviation; please do not use abbreviations if used three or less times.
Response/WHO was deleted.
Line 166 - replace "men and women" with "male and female", because teenagers were included in the study.
Response/ is corrected.
Line 185 - remove the bold in the text.
Response/ bold is removed
The authors